# Oligomerization and Adjuvant Activity of Peptides Derived from the VirB4-like ATPase of *Clostridioides difficile*

**DOI:** 10.3390/biom13061012

**Published:** 2023-06-18

**Authors:** Julya Sorokina, Irina Sokolova, Mariya Majorina, Anastasia Ungur, Vasiliy Troitskiy, Amir Tukhvatulin, Bogdan Melnik, Yury Belyi

**Affiliations:** 1Gamaleya Research Centre for Epidemiology and Microbiology, Moscow 123098, Russia; yv_sorokina@gamaleya.org (J.S.); tabakova@gamaleya.org (I.S.); ungur.nastea@yandex.ru (A.U.); amir_tuhvatulin@yahoo.com (A.T.); 2Institute of Protein Research, Russian Academy of Sciences, Pushchino 142290, Russia; mdudina@phys.protres.ru (M.M.); bmelnik@phys.protres.ru (B.M.); 3Department of Infectious Diseases, Institute of Clinical Medicine, I.M. Sechenov First Moscow State Medical University (Sechenov University), 2 Bolshaya Pirogovskaya St., Moscow 119435, Russia; troickii_vasilii@mail.ru; 4Shemyakin–Ovchinnikov Institute of Bioorganic Chemistry, Pushchino Branch, Russian Academy of Sciences, Pushchino 142290, Russia

**Keywords:** *Clostridioides difficile*, T4SS, VirB4, Walker B, oligomerization, peptide, immune response, adjuvant, ELISA

## Abstract

In a previous study, we demonstrated that the *Clostridioides difficile* VirB4-like ATPase forms oligomers in vitro. In the current investigation, to study the observed phenomenon in more detail, we prepared a library of VirB4-derived peptides (delVirB4s) fused to a carrier maltose-binding protein (MBP). Using gel chromatography and polyacrylamide gel electrophoresis, we found a set of overlapping fragments that contribute most significantly to protein aggregation, which were represented as water-soluble oligomers with molecular masses ranging from ~300 kD to several megadaltons. Membrane filtration experiments, sucrose gradient ultracentrifugation, and dynamic light scattering measurements indicated the size of the soluble complex to be 15–100 nm. It was sufficiently stable to withstand treatment with 1 M urea; however, it dissociated in a 6 M urea solution. As shown by the changes in GFP fluorescence and the circular dichroism spectra, the attachment of the delVirB4 peptide significantly altered the structure of the partner MBP. The immunization of mice with the hybrid consisting of the selected VirB4-derived peptide and MBP, GST, or GFP resulted in increased production of specific antibodies compared to the peptide-free carrier proteins, suggesting significant adjuvant activity of the VirB4 fragment. This feature could be useful for the development of new vaccines, especially in the case of “weak” antigens that are unable to elicit a strong immune response by themselves.

## 1. Introduction

*Clostridioides difficile* [1] is a Gram-positive anaerobic microorganism that causes a wide range of human pathologies (collectively referred to as “*C. difficile* infection”, CDI), including antibiotic-associated diarrhea, pseudomembranous colitis, and toxic megacolon [2]. Studies on the molecular pathogenesis of CDI have shown that the ability of the bacterium to cause disease is largely dependent on the production of toxins [3,4]. However, in-depth studies of the pathogenesis of CDI suggest that other microbial products in addition to toxins may be of value. Such non-toxin molecules contributing to disease progression may include bacterial surface structures [5,6,7,8,9,10,11].

Recently, coding sequences for components of the type 4 secretion system (T4SS) have been described in *C. difficile* genomes [12,13]. In general, the T4SS represents a surface-exposed trans-cell-wall nanomachine capable of transporting protein and DNA effectors into target bacterial or eukaryotic cells, horizontal interbacterial transfer of mobile genetic elements, and DNA exchange with the outside milieu [14,15]. The available data suggest a direct involvement of the system in the pathogenesis of infections caused not only by Gram-negative, but also by Gram-positive, microorganisms [16,17,18]. These data inspired our interest in studying the T4SS of *C. difficile*.

In a recent investigation, we were able to biochemically characterize two major components of the *C. difficile* T4SS: VirB4- and VirD4-like ATPases. An interesting feature of these proteins is the formation of high molecular mass oligomers, which is thought to be mediated by specific fragments of the molecules [19].

It is known that increasing the size of proteinaceous material, which is used as antigenic material during vaccine development, often results in a stronger immune response and represents one of the possible mechanisms of adjuvants [20]. Indeed, it has been shown that particles of sizes between 20 and 200 nm can enter the lymphatic system and be trapped in lymph nodes to be taken up by antigen-presenting cells [21,22]. Under these conditions, a more rapid and more robust immune response is expected, because lymph nodes contain a functionally active population of competent cells and no long-distance migration of immune cells is necessary.

Therefore, in the current study, we focused on the identification of oligomerization-prone peptides in VirB4 and the evaluation of their potential use in protein engineering, specifically as tags with adjuvant activities.

## 2. Materials and Methods

### 2.1. The Materials and Bacterial Strains

Restriction endonucleases, T4 DNA ligase, Phusion DNA polymerase, molecular mass markers, and kits for DNA isolation were obtained from Thermo Fisher Scientific (Moscow, Russia). The lysogeny broth Miller recipe (LB) medium was obtained from Amresco (Solon, OH, USA); liquid chromatography media from GE Healthcare (Moscow, Russia); reagents for agarose and polyacrylamide gel electrophoresis from Bio-Rad (Moscow, Russia); and general laboratory reagents from Merck (Moscow, Russia).

Gene cloning and recombinant protein production were carried out in *Escherichia coli* DH10B and Rosetta (DE3), respectively (Merck). Vectors for cloning and recombinant protein expression in *E. coli* included pMal-c5x (cloning with the NH_2_-terminal maltose-binding protein (MBP), New England Biolabs, Ipswich, MA, USA) and pGEX4T-1 (cloning with the NH_2_-terminal glutathione-S-transferase (GST), GE Healthcare). Anti-MBP antibody (#E8032S) was purchased from New England Biolabs, and anti-FLAG antibody (#14793) from Cell Signaling (Leiden, The Netherlands). The source of the green fluorescent protein coding sequence (eGFP) was the plasmid pEGFP-C, obtained from Clontech (Takara Bio Europe SAS, Saint-Germain-en-Laye, France).

### 2.2. Software Resources

The cloning strategy design and in silico protein analyses were conducted in Vector NTI v.11.0 (Thermo Fisher Scientific). The molecular structure images were constructed by UCSF Chimera [23]. Structural predictions were made using AlphaFold2 (https://colab.research.google.com) and Phyre2 (http://www.sbg.bio.ic.ac.uk/~phyre2) [24,25].

### 2.3. Gene Cloning

For the purpose of cloning the VirB4 gene fragments in-frame with the upstream MBP- or GST-coding sequences, the oligonucleotides shown in Appendix A were used in the PCR. The matrix DNA represented previously described plasmids p994 and p1361, coding for the full-size VirB4 protein (CTn4 locus of *C. difficile* chromosome, NCBI database protein tag ABP89935.1) and its COOH-terminal enzymatic domain (^387^LYYGLN …), respectively [19], as well as the parental genetic constructs. The amplicons were cut with BglII/HindIII, NcoI/SapI (for the MBP-containing constructs), or BamHI/XhoI (for the GST-containing constructs), then ligated into pMal-C5X or pGEX4T-1.

To engineer GFP-coding plasmids, a linker was introduced into the NdeI/NcoI endonuclease sites of p2330 by self-annealing primers #1780/#1781 (Appendix A), thus generating the p2360 molecule with new restriction sites (BamHI, SalI, PstI, EcoRI, and KpnI) in the plasmid. The GFP-coding sequence was PCR-amplified from pEGFP-C with the primers #154/#155 and ligated into p2360 or pMal-C5x using BamHI/EcoRI sites to produce the MBP-tagged GFP protein, with or without the VirB4 oligomerization peptide coded within p2330.

Throughout the text, full-size hybrid proteins and the VirB4-derived peptides are indicated by the ID number of the expression plasmids, with the addition of “p” for the full-size chimeric proteins and “pept” for the specific peptides of VirB4. For instance, the plasmid p2330 codes for the hybrid protein 2330p, consisting of MBP and a specific VirB4-derived peptide 2330pept, while the plasmid p2349 consists of GST and a specific VirB4-derived peptide, 2330pept, etc.

### 2.4. Recombinant Protein Purification

For recombinant protein production, the *E. coli* Rosetta strain was grown in LB medium, supplemented with chloramphenicol at 34 µg/mL plus ampicillin at 100 µg/mL until reaching OD_600_ = 0.5. The induction of expression was performed with 1 mM isopropyl β-D-1-thiogalactopyranoside (IPTG) for 1 h. Bacterial cells of *E. coli* were lysed by sonication, and the resulting extracts were clarified twice at 23,000 g for 20 min. The recombinant proteins were subsequently purified via affinity chromatography using an MBPtrap or GSTtrap column connected to an ÄKTA Explorer liquid chromatography system (GE Healthcare), according to the manufacturer’s instructions. The purified proteins were stored in 10% glycerol/10 mM phosphate buffer with 137 mM NaCl and 2.7 mM KCl (PBS, P4417 Merck) at −20 °C.

To prepare the inclusion bodies, the corresponding producer *E. coli* variants were grown in 40 mL of LB at 37 °C and induced at OD_600_ = 0.5 with 1 mM IPTG for 1 h. Thereafter, cells were pelleted, resuspended in 1 mL of PBS, and lysed by sonication. Unbroken cells were discarded by centrifugation at 1000× *g* for 5 min. IBs were recovered by centrifugation at 23,000× *g* for 20 min. The resulting supernatants were collected as water-soluble fractions (Sol), whereas pelleted IBs were washed sequentially with 1 M NaCl and 1% Triton X-100, then finally dissolved in 1 mL of 6 M urea as the purified preparations. Before the analysis, the samples were diluted 1/10 (for polyacrylamide gel electrophoresis with sodium dodecyl-sulphate (SDS-PAGE)) and 1/50 (for Western blotting with anti-MBP serum).

### 2.5. ATPase Activity

The ATPase activity of the purified proteins was determined using a malachite green phosphate assay (MAK307, Merck). The assay was performed in two steps, as detailed in the user manual. Typically, the reaction mixture in the first step consisted of 0.1 mM ATP, 0.5, or 0.25 µM protein of interest; 2 mM MgCl_2_; 20 mM Tris-HCl; pH = 7.4; and 100 mM KCl for a total volume of 80 µL. The reaction proceeded for 1 h at 35 °C. In the second step, 20 µL of malachite green reagent was added. The results were read on a GloMax-Multi+ plate reader (Promega, Madison WI, USA) following incubation at 22 °C for 30 min. The intensity of the developing green color was proportional to the concentration of phosphate. It was elaborated during ATP hydrolysis, then measured at 600 nm. To convert the optical density into the amount of degraded ATP, a calibration curve was constructed by plotting the optical densities obtained using phosphate standards against the amount of added phosphate [19].

### 2.6. Oligomer Formation Studies

The oligomerization of purified peptides was first studied by gel chromatography on a Superose 6 10/300 column connected to an ÄKTA Explorer liquid chromatography system (GE Healthcare). The proteins, dissolved at concentrations of ~3 mg/mL, were loaded in volumes of 200 µL onto a column equilibrated in PBS, then eluted at 0.5 mL/min. In some experiments, PBS was supplemented with 1 M or 6 M urea in elution buffer. Dextran blue (2000 kD), ferritin (440 kD), aldolase (158 kD), and ovalbumin (43 kD) were used as molecular mass standards.

Analytical ultracentrifugation was used as a second method to probe the oligomerization state of the studied proteins. For this purpose, 60 µL of the protein solution (3 mg/mL) in PBS was loaded onto 90 µL of PBS or 25% sucrose in PBS, then subjected to centrifugation at 393,500× *g* (Hitachi CS150NX micro ultracentrifuge, Tokyo, Japan) at 4 °C for 1 h or 30 min, respectively. Pellets and supernatants were collected separately and analyzed by SDS-PAGE.

Native PAGE (N-PAGE) was the third method used to examine oligomer formation, and was performed in 5% stacking/8% resolution polyacrylamide gels. The buffers and polyacrylamide gel formulations followed those of a Laemmli system [26], but without SDS.

To test the stability of the oligomers in mouse sera, naïve serum was mixed with equal volumes of 2330p oligomers purified by gel chromatography or MBP (both at 0.5 mg/mL), incubated for 1 h on ice, clarified by centrifugation at 23,000× *g* for 20 min at 4 °C, and subjected to Superose 6 chromatography. The fraction samples were subjected to Western blotting with anti-MBP antibody.

### 2.7. Biophysical Methods

Dynamic light scattering (DLS) was measured by the particle size analyzer Malvern Zetasizer Nano ZSP (Malvern, UK) using a 0.3 × 0.3 cm temperature-controlled cuvette and 100 μL of the sample volume. Light scattering was measured at an angle of 173 degrees. The data were analyzed using Malvern software (https://www.malvernpanalytical.com, accessed on 15 June 2023) [27].

The circular dichroism (CD) spectra in the far-ultraviolet region were measured using a J-1500 spectropolarimeter (Jasco, Hachioji, Japan) at a protein concentration of 0.2 mg/mL by means of a 1 mm path quartz cell. Spectra measurements at different temperatures were conducted in 20 mM sodium phosphate buffer at pH 6.2, and were baseline-corrected [28,29]. In order to calculate the percentage of α-helices (%Alpha) in a protein from the ellipticity at 220 nm (Θ_220nm_), the following equation was used: %Alpha = −(Θ_220nm_ + 23400) · 100/30,300 [29].

The tryptophan (Trp) fluorescence of the protein solutions was measured via a Cary Varian spectrofluorometer (Agilent, Santa Clara, CA, USA), using an excitation wavelength of 280 nm and emissions from 300 nm to 450 nm. Measurements were performed in 20 mM sodium phosphate buffer, pH 6.2, at different temperatures using a 10-mm path quartz cell. The protein concentration was 0.2 mg/mL. Changes in the Trp fluorescence spectrum (during heating) were monitored at an intensity of 380 nm (I_380nm_) [27].

### 2.8. General Biochemical Methods

The purified protein preparations were analyzed by SDS-PAGE [26] and Western blotting [30]. The gels were stained with Coomassie Brilliant Blue R-250 (0.25% of the stain in 45.4% methanol and 9.2% acetic acid). Protein concentrations were estimated using Coomassie Brilliant Blue G-250 stain calibrated with bovine serum albumin as a standard [31].

### 2.9. Immunization of Mice

Mice (male, C57 BL/6, 6-week-old, obtained from the animal facility of the Gamaleya Research Centre, housed in conventional microisolator cages, 5 animals in each experimental group) were injected intraperitoneally with 10 µg of purified proteins dissolved in 200 µL of PBS (without any additional adjuvant). One week later, the animals were euthanized by chloroform vapors and blood was taken by heart punctures. The animal research was conducted in compliance with the Animal Welfare Act, as well as other federal statutes and regulations. All animal work was undertaken in strict accordance with the recommendations of the National Standard of the Russian Federation (GOST R 53434-2009). The procedures utilized herein were approved by the Gamaleya Research Center of Epidemiology and Microbiology Institutional Animal Care.

### 2.10. Cytotoxicity Assay

Cytotoxic activity of 2330p was estimated using the VeroE6 (CRL-1586, ATCC) cell line. Cells (2 × 10^5^/mL in 100 µL of DMEM plus 10% bovine fetal serum (Hyclone/Cytiva, Moscow, Russia)) were pre-cultivated in a 96-well tissue culture plate in an atmosphere of 5% CO_2_ at 37 °C for 24 h. Thereafter, we added 10 µL of serial 10-fold dilutions of the tested protein solutions to the wells and proceeded with the incubation for 24 h. The results were obtained by phase-contrast microscopy and by measuring the ATP levels with the CellTiter-Glo^®^ Luminescent Cell Viability Assay (Promega), as described in its instruction manual.

### 2.11. Enzyme-Linked Immunosorbent Assay

The antibody response to purified proteins was investigated using the sera of immunized mice by enzyme-linked immunosorbent assay (ELISA). Proteins (10 µg/mL in 100 µL of 20 mM Tris-HCl buffer, pH = 8.6 with 150 mM NaCl) were incubated overnight at 4 °C in flat-bottom 96-well EIA plates (#224-0096, Bio-Rad, Moscow, Russia). Non-absorbed proteins were aspirated, and the wells were washed 3 times by PBS with 0.05% Tween-20. The well surfaces were blocked with SuperBlock™Tween20 solution (#37536, Thermo) for 1 h at room temperature (RT); washed; and incubated for 1 h at RT with the appropriate mice sera, diluted 1/100 in PBS. Following washings, anti-mouse horseradish peroxidase conjugate (#1705616, Bio-Rad) was added to the wells at 1/1000 dilution in PBS. In the experiments regarding the Ig specificity of the immune response, swine anti-mouse-IgG, -IgM, or -IgA sera (Merck, Mouse Monoclonal Antibody Isotyping Reagents ISO2), as well as the corresponding anti-swine peroxidase conjugate, were used at 1/2000 dilutions. The plates were incubated for 1 h at RT and washed with PBS + Tween, and the reaction was developed with 100 µL of “TMB liquid substrate system” (T0440, Merck) for 10 min or 30 min at RT. The results were read at 450 nm using a GloMax-Multi+ plate reader, following the addition of 100 µL of “Stop reagent for TMB substrate” (S5814, Merck).

### 2.12. Statistical Analysis

The protein purifications and the subsequent protein analyses were performed at least twice for each protein. Groups of 5 mice were used throughout the animal experiments. The results are shown as the means of three measurements, with error bars depicting standard deviations. Student’s *t*-test was used to estimate the statistical significance of the results’ differences.

## 3. Results

VirB4-like ATPase from *C. difficile* forms enzymatically active, high-molecular-mass oligomers [19]. To gain insight into the structural requirements for oligomerization behavior, we subdivided the enzymatic domain of the protein into fragments of different sizes according the Kyte/Doolittle hydrophobicity scale [32], assuming that hydrophobic interactions are the major driving force in protein aggregation [33]. We then expressed and purified VirB4 fragments fused to a well-characterized, highly soluble maltose-binding protein (MBP) [34] and analyzed the resulting VirB4-derived peptide library biochemically (Figure 1).

The resulting MBP::delVirB4 peptide fusions demonstrated different abilities to form oligomers, which were probed by analytical gel chromatography and SDS-PAGE (Figure 2 and Appendix A). Based on these data, we classified the engineered proteins into three major groups. The first group included molecules with high oligomerizing activity (Figure 1, shown in red). Here, full-size water-soluble proteins were eluted almost exclusively in the first peak, suggesting a molecular mass of the formed complex in the range of several megadaltons. The second group included MBP chimeras demonstrating an intermediate level of complex formation (Figure 1, blue); proteins of this type could be found in all major peaks of the chromatograms. The third group included weak oligomerizers (Figure 1, green) and MBP. Their first chromatographic peaks (sometimes elusive) contained negligible amounts of the target product, most of which could be found as soluble monomers in the last peak, suggesting a molecular mass of 50–70 kD, depending on the construct which was analyzed (Figure 2A,B and Appendix A). As shown for the representative samples, their abilities to produce soluble oligomers were proportional to the abilities of the proteins to precipitate and form water-insoluble inclusion bodies (Figure 2C). Our analysis of the selected samples according to the N-PAGE was consistent with the above data and demonstrated oligomer formation by the corresponding proteins, as evidenced by the disappearance of monomeric proteins and the appearance of high-molecular-mass bands in stacking and separation gels (Figure 2D).

We then studied whether the enzymatic activity of the VirB4-like ATPase was related to its ability to form oligomers. Therefore, we expressed and purified a variant of the protein with a deletion of 93 NH_2_-terminal residues. The resulting 2321p protein (Figure 1) showed a dramatic decrease in its ATPase activity in comparison with 1361p (MBP-tagged full-size enzymatic domain of VirB4), despite the similarly high levels of oligomerization (Appendix A). This suggested a lack of interrelationship between the ability of the protein to hydrolyze ATP and to form multimolecular complexes.

An analysis of the engineered peptides allowed us to hypothesize that the central region of the enzymatic domain of VirB4, encompassing amino acid residues 607–657 (i.e., 2330pept) or amino acid residues 615–657 (i.e., 2371pept), contains structural features most significantly contributing to the in vitro oligomerization behavior of the engineered proteins (Figure 1). We investigated this phenomenon in more detail using biochemical methods and found that analytical high-speed centrifugation in PBS resulted in complete pelleting of the protein, whereas sucrose gradient sedimentation allowed a portion of the protein to remain in the supernatant (Figure 3A). The cytosolic high-molecular-mass complex formed by 2330p was stable in the presence of 1M urea solution, but disintegrated in 6 M urea (Figure 3B–D). This observation suggested quite strong intermolecular interactions. The formed oligomers were also quite stable in the mouse body fluid (blood serum) (Appendix A). The protein easily passed through a membrane filter with a pore diameter of 0.2 µm (Spartan 13/0.2RC, GE Healthcare) (Appendix A). Neither the substitution of MBP by GST (2349p), the addition of a COOH-terminal FLAG tag (2359p and 2378p), nor the insertion of GFP between MBP and 2330pept (2361p) altered the oligomerization capacity of these newly engineered molecules (Figure 4, Appendix A).

Next, we looked into structural reorganization of the aggregating proteins by using GFP, the fluorescence activity of which depends on the correct folding state [35]. The MBP::GFP chimeric protein lacking the delVirB4 fragment (construct 2363p) behaved as a monomer in gel chromatography experiments. The measurement of a fluorescent signal in gel chromatography fractions showed that the MBP::GFP hybrid was highly active, and the peak of its fluorescent activity coincided with the expected monomeric form of the protein. In contrast, the fluorescent activity of oligomerization-competent 2361p was greatly reduced, and only residual activity was found in fractions where monomers were expected to elute (Figure 4A). These results pointed toward a significant loss of native structure in GFP due to its tagging by the oligomerization-prone VirB4-derived peptide.

The “unfolding” influence of 2330pept upon the partner protein was further studied by spectroscopic methods (Figure 5). As seen from DLS experiments shown in Figure 5A,B, MBP behaved as a monomer at all tested temperatures. Higher temperatures resulted in partial unfolding of the protein, which was reflected by an increase in particle size from 5–6 nm at 30 °C to 8–9 nm at 70 °C. In contrast, the MBP::2330pept chimera formed oligomers at 30 °C, a substantial fraction of which represented low-ordered complexes (tetramers–pentamers) of 15–20 nm as well as higher-ordered structures of 30–40 nm, while heating to 60 °C and 70 °C produced particles of ~100 nm and ~1000 nm, respectively.

The shapes of the spectra shown in Figure 5C indicate that both proteins had a predominantly α-spiral structure at 20° C. Heating to 90 °C resulted in almost complete unfolding of MBP, while for 2330p, a considerable amount of a secondary structure remained intact. This can be seen both by the shape of the CD spectra (Figure 5C) and by the value of ellipticity at 220 nm (Figure 5D). The melting data of the proteins investigated by CD (Figure 5D) and Trp fluorescence (Figure 5E) agreed quite well, demonstrating that the major denaturation of 2330p occurred at about 50 °C, while the complete unfolding of MBP occurred at about 60 °C.

To estimate the percentage of α-helices, we used the equation described in [29]; this method allowed us to calculate the percentage of α-helices from the value of ellipticity at 220 nm. The α-helix content of MBP and 2330p at 20 °C was approximately the same, i.e., 15% and 18%, respectively. When the temperature was increased to 90 °C, the α-helix content in MBP dropped down to 2%, i.e., the number of α-helices decreased by more than 7-fold. This is consistent with the assumption of almost complete unfolding of heated MBP. However, when protein 2330p was heated to 90 °C, about 9% of the α-helices remained intact, i.e., the number of α-helices decreased by only half.

The ability of 2330p to form oligomers suggested that 2330pept could be used as a scaffold in peptide engineering when a protein with a very high molecular mass is particularly desirable, for example, in the construction of highly immunogenic protein variants. To test whether the VirB4-derived peptide encoded by p2330 possesses adjuvant activity, we first confirmed its safety in cell cultures (Appendix A). We then immunized mice with several recombinant chimeric protein variants and tested the obtained sera for specific antibody levels (Figure 6).

As can be seen from the results, the peptide 2330pept significantly increased antibody levels against companion proteins after immunization. The most impressive data were obtained with the MBP hybrids. Here, immunization with MBP as a single molecule failed to induce antibody production, but the attachment of the VirB4 fragment boosted MBP-specific antibody production of the IgA, IgG, and IgM types (Figure 6A,D). Similarly, the same *C. difficile*-derived peptide enhanced the immune response to other antigens (GST and GFP), although the response to these chimeras was variable (Figure 6B,C). Importantly, experiments with cross-reactivity (i.e., reaction of sera against MBP-containing hybrids with GST-containing constructs absorbed on plastic, and vice versa) suggested a low level of immunogenicity of the tested VirB4 peptide per se (Appendix A).

## 4. Discussion

The formation of functionally competent oligomers from VirB4 proteins represents an important feature of T4SS ATPases [36]. In line with the published data [37,38,39], our recent study of the enzymatic domain of the VirB-like protein from *C. difficile* demonstrated almost 100% of its oligomerization rate [19]. We decided to refine fragments of this protein which possessed aggregation activity, keeping in mind their possible usage in protein engineering as oligomerization-prone tags.

In these studies, we were able to reduce the size of the peptide with the highest oligomerization activity down to 40 amino acid residues (Figure 1, 2371pept). Its NH_2_-terminus represents a sequence, known as the Walker B motif (WB, the consensus sequence is R/KxxxGxxxLhhhDE, where “h” stands for hydrophobic and “x” stands for any other amino acid residue [40]). This motif represents one of the phylogenetically conserved ATP-binding domains, and is characteristic of several ATPases [41]. Bearing in mind the ubiquity of WB among different ATP-binding proteins, it would be interesting to study the oligomerization activity of WB-containing peptides obtained from different sources in the future, including *C. difficile* VirD4-like ATPase. As predicted by AlphaFold2, the latter molecule is structurally highly homologous to VirB4, despite considerable primary sequence dissimilarities (Figure 7 and Appendix A).

The available crystal structure of a VirB4-related protein from *Thermoanaerobacter pseudethanolicus* [42] and structural modeling using AlphaFold2 [25] allowed us to model the position of 2371pept within the VirB4 molecule of *C. difficile* with good confidence (the predicted local Distance Difference Test (lDDT) being between 70% and 90%) (Appendix A). While the Walker B motif sequence traverses the enzymatic domain of VirB4 of *C. difficile* and is located only partially on its surface (Figure 7B, red), the COOH-terminal stretch of amino acid residues, converting 2371p into a strong oligomerizer, is located fully on the surface (Figure 7B, yellow. It can, theoretically, participate in intermolecular contacts, leading to the formation of high-molecular-mass homocomplexes.

In our investigations, WB that was lacking the COOH-terminal amino acid residues of an efficient oligomerizer 2371pept failed to promote its aggregation (Figure 1, 2373pept, green, Appendix A). This fact suggests that direct interaction of adjacent WB motifs located in adjacent molecules plays no direct role in complex formation per se. In 2371pept, however, WB might provide a backbone necessary for optimal orientation of its oligomerization-competent COOH-terminal extension. Thus, both subdomains could be necessary for aggregation of the protein. Consistent with this speculation are the experimental data showing that splitting the strong oligomerizer 2330pept into 2331pept and 2335pept, or into 2370pept and 2335pept, resulted in a decrease in complex formation by the resulting fragment molecules (Figure 1).

The important role of structural features in the aggregation of molecules during inclusion body formation is well known. Microscopically, bacterial IBs exhibit an amorphous morphology [43]. However, data from Fourier transform infrared spectroscopy (FTIR) have provided evidence that IB fractions are capable of exhibiting beta-sheet amyloid-like or native-like conformations [44]. In the case of hybrids consisting of an aggregation-prone peptide (e.g., Alzheimer’s Aβ43) and GFP, the resulting aggregates retain fluorescent activity, suggesting that GFP molecules can maintain native-like folding states [45]. However, in our experiments, GFP-containing chimeras lost their fluorescence activity. Moreover, the spectroscopic methods utilized herein clearly indicated that the attachment of the VirB4-derived oligomerization-prone peptide significantly altered the structural characteristics of the partner MBP molecule.

The biochemical mechanisms of complex formation by 2330p and 2371p are not completely clear to us. Comparison of the properties of the delVirB4 peptides engineered in our study demonstrated that neither size, nor pI, nor hydrophobicity level could predict complex formation by the molecules (Appendix A). In fact, for these characteristics, both 2330pept and 2371pept were in the middle of the spectrum for each parameter, and did not differ significantly from the weak oligomerizers. As was deduced from the data obtained in our experiments, we speculate that the process of oligomerization may be initiated by the intermolecular interaction of oligomerization-prone peptides, and further proceeded by the intermolecular interaction of the partially unfolded partner proteins. In the latter case, the functions of the delVirB4 peptides may also include changing the structure of the partner proteins.

Aggregation of recombinantly produced proteins is a process often seen in biotechnology investigations [46]. Initially, proteins in IB were considered to be a waste “by-product” of the bacterial cell factories during recombinant overproduction [47]. However, it soon became obvious that some recombinant proteins in IB could retain a native-like secondary structure and biological activity, making them a “treasure trove” of bioactive products [46,47,48]. On the other hand, protein aggregation has been shown to alter the immunogenicity of biotherapeutics, leading to immune-mediated side effects such as autoimmunity, which can directly block their therapeutic function or cause autoallergy. [49].

In the published studies, the utilized IBs were produced in bacteria as water-insoluble materials. In contrast, we decided to attempt to use oligomerization-prone peptides as nucleation centers for the production of *water-soluble* high-molecular-mass complexes. Such a task would result in aggregates of smaller sizes and could be important in, e.g., immunological investigations as a means to increase the immunogenic properties of different antigens by oligomerization.

Adjuvants represent critically important components of vaccines, and are necessary for enhancing their protective efficacy by specifically boosting antibody generation against specific antigens. However, adjuvant discovery seems to be one of the slowest processes in the pharmaceutical industry. Since the introduction of insoluble aluminum salts (alum) as the first adjuvants more than 100 years ago, only ten molecules have been approved for use as components of vaccines against infectious diseases [20].

On the basis of their functions, adjuvants can be divided into several groups, among which the substances affecting the delivery of the partner antigens (the so-called “delivery systems”, such as alum, lipid nanoparticles, etc.) represent the most studied cluster [20]. Their mechanism can be explained, at least in part, by the increased size of the antigenic material. The peptide described in our investigation fits this mechanism quite well. The oligomers which are formed are expected to have molecular masses ranging from 300 kD to several megadaltons. As seen from the experiments using dynamic light scattering, membrane filtration and sucrose ultracentrifugation (the protocol we used for the latter method is routinely applied for the isolation of crude ribosomes [50]), the size of the formed homocomplex can reach the size of a cellular organelle (i.e., 20–30 nm), but must be smaller than 200 nm (the cut-off diameter of the filter membrane).

Peptides that induce oligomer formation and are conjugated to the target antigen by gene cloning are known in the literature and represent good candidates for use in vaccine development [51,52,53]. One of the model molecules in these investigations represents the complement inhibitor C4-binding protein (C4bp), which forms heptamers when fused to the protective antigens [54,55]. However, our study differs significantly from previous work published by other groups. First, the molecule which we identified produced oligomers with a wide mass range, but those within the size limits seemed to be optimal for immune recognition. Second, the peptide used in our study affected the structure of the partner protein, as seen with the GFP-containing chimeras and from our spectroscopic data. The significance of this effect is not clear at this time, but it is worth investigating. We speculate that changes in the antigenic landscape of the partner molecule could lead to an enhanced immune response, especially against weak antigens tolerated by an organism’s immune system. Third, the oligomerizing peptides similar to the one we have identified may be part of a large family. Indeed, Walker B motifs and related sequences are widely distributed in different proteins. It would be interesting to learn in the future how structural peculiarities would affect the oligomerization and adjuvant activities of these molecules.

It should be noted that the peptide (2330pept) described in the current paper does not represent a molecule with universal adjuvant activity. In fact, as seen in our experiments, the antibody responses to the hybrid molecules containing MBP, GST, or GFP were subjected to significant variation. As mentioned above in the Discussion section, the relationship between aggregation and immunogenicity is a scientific topic with a long and successful history. However, we believe that the biomolecules described in this study are promising and interesting from both basic and applied science perspectives. With our data in hand, future bioengineering efforts aimed at constructing vaccines suitable for human prophylaxis against serious infectious diseases, including CDI, may be fruitful.

We acknowledge the existence of obvious limitations of our study, which should be addressed in the future by experiments more carefully concerning the immune response. Specifically, the experimental design should implement a variety of modern immunological methods with the important goal of estimating the main parameters of the protective activity, including its duration, the role of different immune cell populations, and immunity power.

## Figures and Tables

**Figure 1 biomolecules-13-01012-f001:**
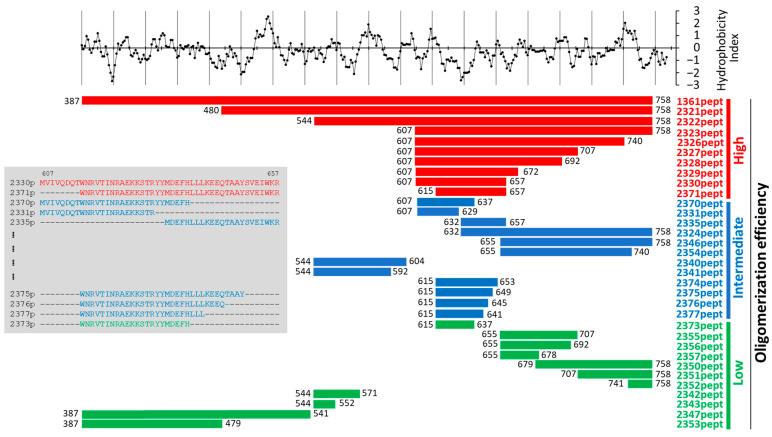
VirB4-derived peptides and their oligomerization behavior. Upper graph represents hydrophobicity scale following the Kyte/Doolittle protocol (https://web.expasy.org/protscale/, accessed on 15 June 2023) [32]. Each dot above zero represents hydrophobic amino acid residue; below zero, hydrophilic. The protein 1361pept is an enzymatic domain of the ATPase. The engineered peptides are shown as thick horizontal lines, and are grouped according to the oligomerization level (see the Results section)—high (red), intermediate (blue), and low (green). Numbers before and after each line show the starting and ending amino acid residues in relation to the full-size proteins. The grey insert shows sequence details on the selected most interesting peptides.

**Figure 2 biomolecules-13-01012-f002:**
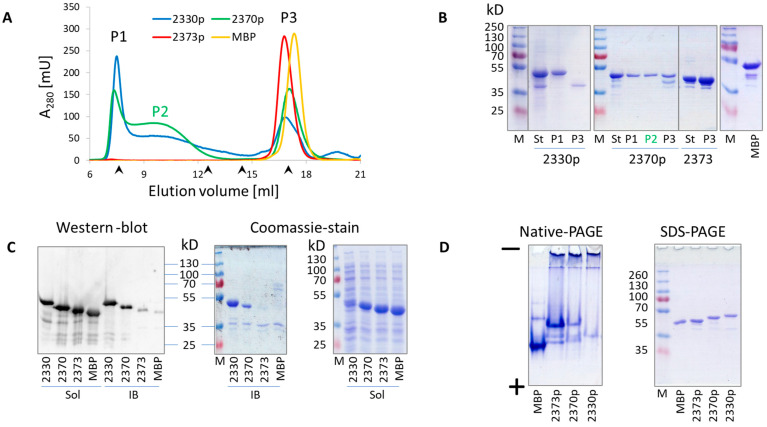
Oligomerization profiles of the representative VirB4-derived proteins. (**A**) Superose 6 analytical chromatography of the representative proteins. Arrowheads on the X-axis indicate elution volumes of molecular mass markers (from left to right: 2000 kD Dextran Blue, 440 kD Ferritin, 158 kD Aldolase, and 43 kD ovalbumin). Fractions from the peaks (P1, P2, P3) were analyzed, as shown in (**B**). If the peak was observed with a single protein, it was marked by the corresponding line color. The X-axis represents elution volume in ml, while the Y-axis shows the optical density of the solution at 280 nm. (**B**) SDS-PAGE analysis of the gel chromatography fractions. M, molecular mass marker; St, starting material (proteins purified by MBPtrap chromatography); MBP, maltose-binding protein. (**C**) Western blot analysis with anti-MBP antibody (left panel) and SDS-PAGE analysis (center and right panels) of the water-soluble (sol) and IB fractions obtained from the corresponding protein-producing *E. coli* cultures. (**D**) N-PAGE and SDS-PAGE of the representative proteins, purified by MBPtrap chromatography.

**Figure 3 biomolecules-13-01012-f003:**
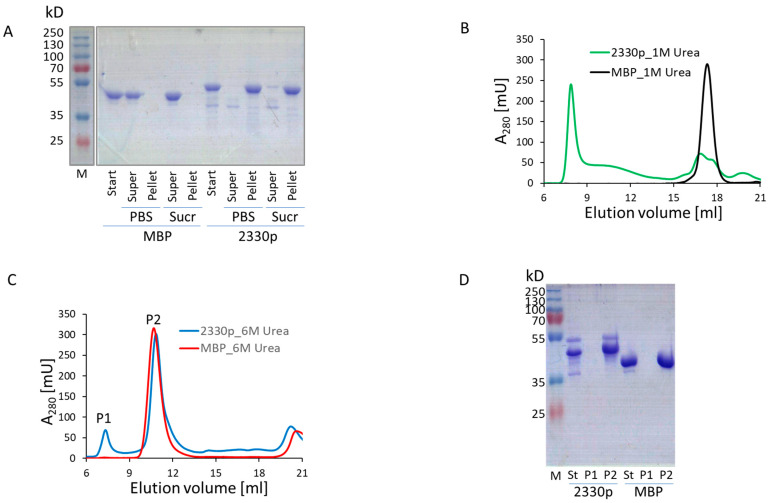
Biochemical properties of MBP::2330pept chimera. (**A**) SDS-PAGE analysis of fractions obtained by ultracentrifugation in PBS or sucrose gradient (Sucr). Start, starting material (proteins purified by the MBPtrap chromatography); super, supernatant after ultracentrifugation; pellet, pellet fraction after ultracentrifugation; M, molecular mass markers. (**B**) Analytical gel chromatography on Superose 6, equilibrated in PBS with 1 M urea. (**C**) Analytical gel chromatography on Superose 6, equilibrated in 6 M urea. (**B**,**C**) The X-axis represents the elution volume in ml, while the Y-axis shows the optical density of the solution at 280 nm. (**D**) SDS-PAGE analysis of fractions from the main peaks of the chromatogram shown in (**C**). St, starting material (proteins purified by the MBPtrap chromatography).

**Figure 4 biomolecules-13-01012-f004:**
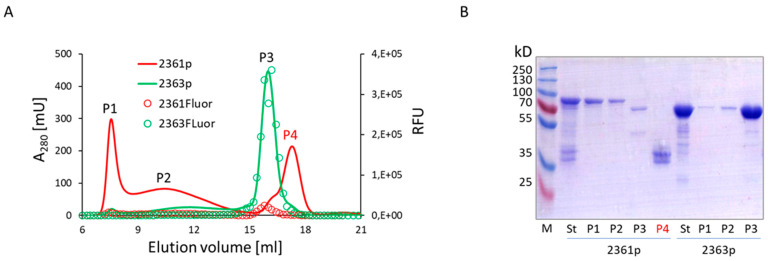
Biochemical properties of MBP::GFP and MBP::GFP::2330pept chimeras. (**A**) Analytical gel chromatography on Superose 6. The optical density at 280 nm (solid lines) and the fluorescence (open circles) of the corresponding proteins are shown. RFU, relative fluorescence units. If the peak is seen only with a single protein, it is marked by the corresponding line color. The X-axis represents the elution volume in ml, while the Y-axis shows the optical density of the solution at 280 nm or the fluorescence level, measured in RFUs. (**B**) SDS-PAGE analysis of the fractions from the main peaks shown in panel (**A**). M, molecular mass markers; St, starting material (proteins purified by the MBPtrap chromatography).

**Figure 5 biomolecules-13-01012-f005:**
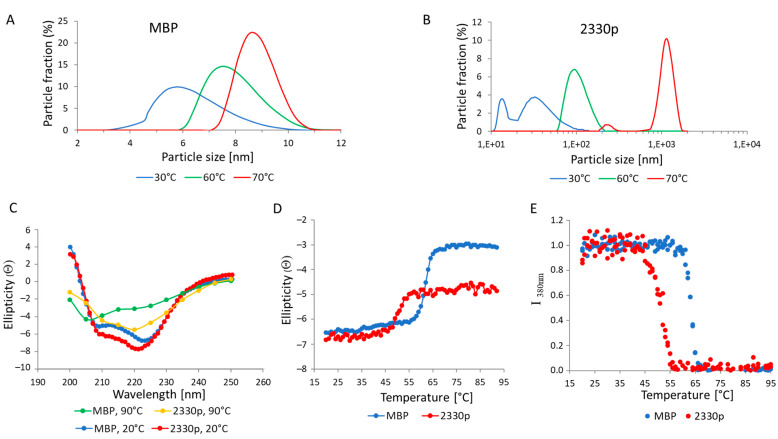
Structural properties of MBP and MBP-tagged 2330pept. Size distribution of MBP (panel **A**) and MBP::2330pept (panel **B**) measured by dynamic light scattering at different temperatures. The X-axis represents particle sizes measured by light scattering, while the Y-axis shows a percentage of the particles of a selected size. (**C**) CD spectra of MBP and MBP::2330pept, taken at 20 °C and 9 °C. (**D**) CD spectra of MBP and MBP::2330pept, taken at 220 nm and at different temperatures. Ellipticity (Θ) is shown on the Y axis in deg·cm^2^·dmol^−1^·10^−3^. (**E**) Tryptophan fluorescence emission intensity (I_380nm_), measured at 380 nm at different temperatures.

**Figure 6 biomolecules-13-01012-f006:**
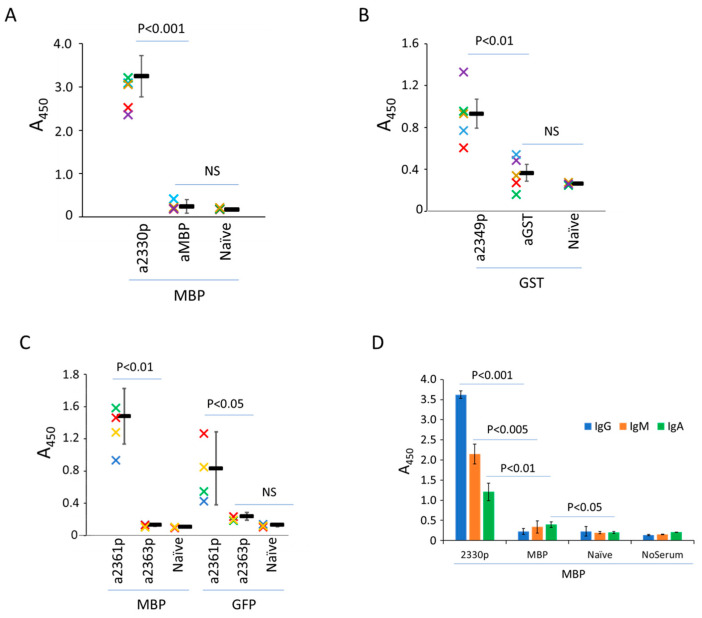
Enzyme-linked immunosorbent assay of mice sera. The animals were immunized with the following proteins: MBP, MBP::2330pept (2330p), GST, GST::2330pept (2349p), MBP::GFP (2363p), and MBP::GFP::2330pept (2361p). The sera were tested on plates sensitized with MBP (panel **A**,**D**), GST (Panel **B**), MBP, or GFP (panel **C**). The reactions were developed either with the total anti-mouse conjugate (**A**–**C**) or with specific anti-A, anti-G, or anti-M antibodies (panel **D**, see Materials and Methods for details). Naïve control sera of non-immunized animals. Data are shown for the individual sera (colored crosses) and as means and standard deviations. The types of mouse sera are shown on the X-axis. The Y-axis shows the optical density of the reaction, measured at 450 nm. The difference between the groups was calculated according to Student’s t-test as either non-significant (NS) or significant at *p* < 0.001, *p* < 0.01, or *p* < 0.05.

**Figure 7 biomolecules-13-01012-f007:**
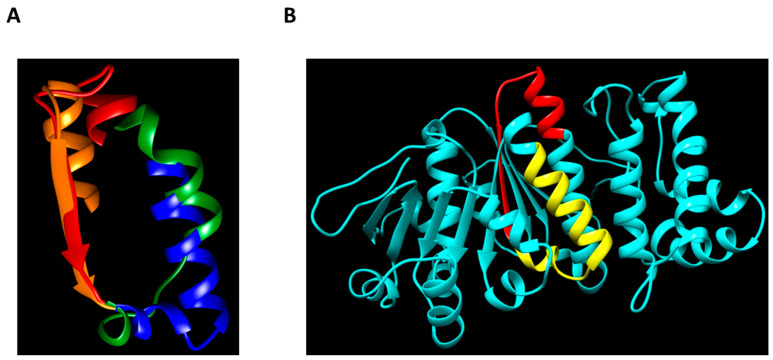
Predicted structures of *C. difficile* T4SS ATPases. (**A**) Partial alignment of the VirB4 and VirD4 structures from *C. difficile*. The predicted Walkers B are shown in red (VirB4) and orange (VirD4), while their COOH-terminal extensions are shown in blue (VirB4) and green (VirD4). (**B**) AlphaFold2 structure prediction of the VirB4-like protein from *C. difficile*. The smallest oligomerization-prone peptide is shown in red (Walker B) and yellow (COOH-terminal extension).

## Data Availability

The plasmids coding for the proteins used in the study are available upon request from Y.B.

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
