# Peer review of "Oligomerization and Adjuvant Activity of Peptides Derived from the VirB4-like ATPase of Clostridioides difficile"

_biomolecules, 2023, doi:10.3390/biom13061012_

Round 1
Reviewer 1 Report (Previous Reviewer 1)
The revised manuscript by Sorokina et al. describes the effect of Vir-B4-derived peptides attached by genetic engineering to the C terminus of three model proteins (GFP, MBP, and GST) on their oligomerization and immune response.
Though the authors characterized the effect of several peptides on the oligomerization of the model protein, I feel that the study is in a very preliminary state. The scientific/technical significance of this study is still hard to understand.
1. Other groups have reported similar results with different model proteins and peptides, but there are few if any, mentions of these independent studies. Thus, the author's observations are not novel and barely add to the present status of knowledge from an aggregation/immunogenicity viewpoint.
In the revised version, the authors added some references and the corresponding discussion. However, recent and essential references are still missing and should be included for the sake of completeness and an insightful discussion. Especially the following reports provide a detailed analysis and an in-deep discussion of the relationship between protein oligomerization/aggregation and the immune response(e.g.:Kibria Md.Golam et al. European Journal of Pharmaceutics and Biopharmaceutics. 2021 Aug 165:13-21) Pham NB, Meng WS. Protein aggregation and immunogenicity of biotherapeutics. Int J Pharm. 2020; Lundahl et al, RSC Chem Biol. 2021 May 4;2(4):1004-1020.)
2. In addition, the characterization of both the aggregational properties and immunogenicity is poor. In addition to SEC and SDS page, the authors could have performed state-of-the-art spectroscopic measurements to characterize the conformation and the nature of the aggregates.
In the revised version, the authors have performed Cd and DLS. They could also perform ANS analysis and Thioflavin T analysis to characterize the aggregates' nature further.
3. Similarly, the immune response is poorly characterized. Even from a technical viewpoint, it is unclear how many mice were used and if adjuvants were used (perhaps they did not use adjuvant).
In the revised version, the authors have a better description of the condition. The authors should report the titer and the dilution data rather than the OD in Fig 6. The OD can be shown in the supplemental if the authors wish to do so, but the titer is more reliable for monitoring the immune response. In addition, the authors should discuss the limitation of their research in terms of immune response characterization (No FACS, no long-term study, no neutralization assay): See, for example, , Islam MM et al., Front Immunol. 2020 Mar 17;11:333. Ohkuri et al, J Immunol. 2010 Oct 1;185(7):4199-205. doi: 10.4049/jimmunol.0902249
4. To conclude, this manuscript looks more like a case report or a well-written Bachelor's thesis. I believe that much more improvement and a better focus on the significance of the research is desirable.
The authors have improved their manuscript, but I would like to emphasize that the relationship between aggregation and immunogenicity has been studied in detail (see the above references). The phenomenon of aggregation/immunogenicity is thus not a new finding, but the current data are worth reporting as they apply to a specific case. Thus the authors may want to discuss the limitations of their study and emphasize the significance of their findings in the more specific framework of biotechnological application against Clostridiodis difficile.

Author Response
We thank the distinguished reviewer for the critical reading of our revised manuscript. Our responses to the valuable comments are provided below.
Comment 1. “In the revised version, the authors added some references and the corresponding discussion. However, recent and essential references are still missing and should be included for the sake of completeness and an insightful discussion. Especially the following reports provide a detailed analysis and an in-deep discussion of the relationship between protein oligomerization/aggregation and the immune response (e.g.:Kibria Md.Golam et al. European Journal of Pharmaceutics and Biopharmaceutics. 2021 Aug 165:13-21) Pham NB, Meng WS. Protein aggregation and immunogenicity of biotherapeutics. Int J Pharm. 2020; Lundahl et al, RSC Chem Biol. 2021 May 4;2(4):1004-1020.)”.
Response. We agree that these are two nice papers devoted to the relationship between immunogen particle size and specific immune response. However, in the paper by Dr. Kibria et al, the physicochemical properties of the particles formed by the VHH antibodies studied differed significantly from those formed in our experiments. As shown in Fig. 2, in contrast to our data, the material used for immunization was completely unfolded even at 37°C and the size of the aggregates formed was in the range of 1 µm (dashed lines on the graphs). Therefore, we prefer not to cite this paper in our manuscript. The second paper by Dr. Lundahl et al. is a review article and it fits perfectly to the topic of our manuscript. We gratefully include this reference (lines 423-426).
Comment 2. “In the revised version, the authors have performed Cd and DLS. They could also perform ANS analysis and Thioflavin T analysis to characterize the aggregates' nature further”.
Response. Thank you for your suggestions. These methods are indeed good ways to move forward. However, from our point of view, the CD, DLS and Trp fluorescence we use are sufficiently informative at the current stage of our research and we prefer not to go deep into molecular biophysical studies. Instead, we are currently investigating the range of antigenic changes induced by the VirB4 peptide. These studies will be the subject of a separate paper.
Comment 3. “In the revised version, the authors have a better description of the condition. The authors should report the titer and the dilution data rather than the OD in Fig 6. The OD can be shown in the supplemental if the authors wish to do so, but the titer is more reliable for monitoring the immune response. In addition, the authors should discuss the limitation of their research in terms of immune response characterization (No FACS, no long-term study, no neutralization assay): See, for example, Islam MM et al., Front Immunol. 2020 Mar 17;11:333. Ohkuri et al, J Immunol. 2010 Oct 1;185(7):4199-205. doi: 10.4049/jimmunol.0902249”.
Response. Indeed, we have presented our ELISA data by means of optical density measurements. However, I would like to mention that there are different ways of presenting ELISA data (titrations, as mentioned by the Reviewer (e.g., doi.org/10.3390/molecules28114567); %% of negative control (e.g., doi: 10.3389/fimmu.2023.1183825); by using COI (cutoff index value, e.g., doi.org/10.1016/j.jiph.2023.05.036); used by us OD measurements (e.g., doi.org/10.1371/journal. pone.0287107) etc. The use of each method obviously depends on different subjective and objective conditions. The method we use is probably the simplest, but the difference between the control groups and the 2330pept fused proteins is clearly visible. From this point of view, our method works. However, due to the fact that this point has attracted special attention from the distinguished reviewer, we agree that we should be more careful in choosing the type of ELISA data presentation in future investigations. In addition, as suggested, we have added more discussion of the limitations of our research in terms of the immune response (lines 476-481).
Comment 4. “The authors have improved their manuscript, but I would like to emphasize that the relationship between aggregation and immunogenicity has been studied in detail (see the above references). The phenomenon of aggregation/immunogenicity is thus not a new finding, but the current data are worth reporting as they apply to a specific case. Thus the authors may want to discuss the limitations of their study and emphasize the significance of their findings in the more specific framework of biotechnological application against Clostridiodis difficile”.
Response. We have done this, please see lines 470-475. Hope, this is fine.
Reviewer 2 Report (Previous Reviewer 3)
I was one of the reviewers for the earlier submission of the manuscript. Looking at the revised version, I find the authors have done necessary corrections, hence there there should be no reason for me add anything more.
Author Response
We would like to thank the distinguished reviewer for her/his attempts to improve the manuscript.
Reviewer 3 Report (New Reviewer)
This is a very sound work. In this manuscript, the author prepared a library of VirB4-derived peptides from VirB4-like ATPase. The peptide library was then fused to MBP for bacteria expression. Using SDS-PAGE, DLS, and CD, the author found 2330P is the key component for oligomerization. Using mice model, the author demonstrated 2330P fusion can significantly increase the adjuvant activity. The discovery of 2330P peptide can be optentially used fro "weak" antigens to elicit a strong immune repsonse.
I would recommend to accept this paper in current form.
Author Response
We would like to thank the distinguished reviewer for her/his attempts to improve the manuscript.
This manuscript is a resubmission of an earlier submission. The following is a list of the peer review reports and author responses from that submission.
Round 1
Reviewer 1 Report
This manuscript by Sorokina et al. describes the effect of Vir-B4-derived peptides attached by genetic engineering to the C terminus of three model proteins (GFP, MBP, and GST) on their oligomerization and immune response.
Though the authors characterized the effect of several peptides on the oligomerization of the model protein, I feel that the study is in a very preliminary state. The scientific/technical significance of this study is particularly hard to understand.
The results (including the supplementary data) indicate that except for a few peptides, adding the peptide did induce oligomerization of MBP (there are few experiments with GST. The second finding is that the attachment of peptide 2330 to MBP and, to a lesser extent, to GST increased their immunogenicity (probably in terms of IgG or IgM titer, but this is not indicated).
Other groups have reported similar results with different model proteins and peptides, but there are few, if any, mentions of these independent studies. I would thus say that the author’s observations are not novel and barely add to the present status of knowledge.
In addition, the characterization of both the aggregational properties and immunogenicity is poor. In addition to SEC and SDS page, the authors could have performed state-of-the-art spectroscopic measurements to characterize the conformation and the nature of the aggregates. Similarly, the immune response is poorly characterized. Even from a technical viewpoint, it is unclear how many mice were used and if adjuvants were used (perhaps they did not use adjuvant).
I believe that much more improvement and a better focus on the significance of the research are desirable.
Reviewer 2 Report
This is a basic research for the characteristic of VirB4-like ATPase of Clostridioides difficile. This paper determined the oligomer formation, enzymatic activity and adjuvant activity of the selected peptides.
There are several points that need to be addressed.
- The description of the significance and originality of the research needs to be strengthened in the Introduction section.
- In Figure 5, the experimental design lacks a control group of the administration of protein 2330p alone. Whether administration of 2330p alone can induce antibody titer or not? Please justify the number of animals in each treatment group as appropriate.
- Did the authors evaluate the safety of this purified protein? For example, to determine the cytotoxicity of 2330p on mammalian cells or provide safety data from animals.
- The labels on Y-axis and X-axis are not described clearly. The authors use the abbreviation in several figures without brief description. What is the meaning of RFU and A450?
Reviewer 3 Report
The manuscript, biomolecules 2170341 - Oligomerization and adjuvant activity of peptides derived from the VirB4-like ATPase of Clostridioides difficile, by J Sorokina et al investigate the aggregation potential of VirB4-derived peptides. The authors utilize gel filtration chromatography, polyacrylamide gel electrophoresis, and immunoblotting to identify aggregation-prone regions. A fusion tag is included with the peptide to facilitate chromatography-based separation in addition to utilizing this as an antigen upon injection into mice. Overall, the authors suggest the VirB4 peptides, given their aggregation capability, could be utilized as a promising adjuvant for a connected partner protein of interest; thereby highlighting protein engineering prospects and the clinical applicability of their findings. Yet there are several areas that need improvement. The reviewer has the following questions, comments, and suggestions for the authors.
1) Similar to what was reported by the authors earlier (in J Bacteriol. 2021; 203(21): e00359-21), the current study, besides understanding aggregation-prone peptides, should also attempt to characterize the biophysical nature of the oligomers formed by peptides using electron microscopy or AFM, and validated by circular dichroism or FTIR for secondary structural knowledge.
2) Does ViB4-MBP or -GST or -GFP aggregate in vivo upon injection? Any response to this question should be supported with experimental data.
3) Explain the significance of the results that show the aggregates dissolve at a high concentration of Urea (6M) compared to a lower urea concentration (1M).
4) Despite the ability of the oligomer-prone peptide to perform as an adjuvant, it is able to negatively influence the functional activity of the partner protein (GFP fluorescence in this study). Please highlight briefly (in the Discussion section) what necessary modifications would be needed, from a protein engineering standpoint, to fix this problem.
5) What are the authors' views on the antigenicity of the untagged or epitope (MBP or GFP)- tagged bacterial peptides themselves when injected in a live animal? Include a succinct description under the Discussion section.
6) Please ensure the correctness of all references cited.